# Changes of Bioactive Components and Antioxidant Capacity of Pear Ferment in Simulated Gastrointestinal Digestion In Vitro

**DOI:** 10.3390/foods12061211

**Published:** 2023-03-13

**Authors:** Xiaoying Zhang, Yiming Li, Yue Li, Jiangli Zhao, Yudou Cheng, Yongxia Wang, Junfeng Guan

**Affiliations:** 1Institute of Biotechnology and Food Science, Hebei Academy of Agricultural and Forestry Sciences, Shijiazhuang 050050, China; 2College of Life Sciences and Food Engineering, Hebei University of Engineering, Handan 056000, China

**Keywords:** pear ferment, *in vitro* simulated gastrointestinal digestion, bioactive components, antioxidant capacity

## Abstract

Fruit ferment is rich in polyphenols, organic acids, enzymes, and other bioactive components, which contribute to their antioxidant ability. In this study, we investigated the effect of the simulated gastric and intestinal digestion *in vitro* on the total phenolic content (TPC), total flavonoid content (TFC), phenolic components content, organic acid content, protease activity, superoxide dismutase (SOD) activity, 1,1-diphenyl-2-picrylhydrazyl (DPPH) radical scavenging activity (DPPH-RSA), hydroxyl (·OH) radical scavenging activity (·OH-RSA), and total reducing capacity in ‘Xuehua’ pear (*Pyrus bretschneideri* Rehd) ferment. The result showed that the TPC, TFC, protease activity, and phenolic components such as arbutin, protocatechuic acid, malic acid, and acetic acid showed a rising trend during the simulated gastric digestion in ‘Xuehua’ pear ferment, and these components might contribute to the increasing of ·OH-RSA and total reducing capacity. The SOD activity and epicatechin content showed an increasing trend at first and then a decreasing trend, which was likely associated with DPPH-RSA. During *in vitro*-simulated intestinal digestion, the majority of evaluated items reduced, except for protease activity, quercetin, and tartaric acid. The reason for the decreasing of bio-accessibility resulted from the inhibition of the digestive environment, and the transformation between substances, such as the conversion of hyperoside to quercetin. The correlation analysis indicated that the antioxidant capacity of ‘Xuehua’ pear ferment was mainly affected by its bioactive compounds and enzymes activity as well as the food matrices and digestive environment. The comparison between the digestive group with and without enzymes suggested that the simulated gastrointestinal digestion could boost the release and delay the degradation of phenolic components, flavonoids, and organic acid, protect protease and SOD activity, and stabilize DPPH-RSA, ·OH-RSA, and total reducing capacity in ‘Xuehua’ pear ferment; thus, the ‘Xuehua’ pear ferment could be considered as an easily digestible food.

## 1. Introduction

Edible plant ferments are kinds of functional food fermented from fruits, vegetables, cereals, and herbs by microorganisms such as yeast and *lactobacillus* [1,2,3]. Ferments have various health benefits such as antioxidant activity [4], detoxification of liver [5], laxative effect [6], immunity improvement [7], and prevention of certain chronic diseases [8,9,10].

The bioactive component activities in ferments are bound to change after various biochemical reactions through human gastrointestinal digestion, such as transformation and degradation [11], which in turn affect their bioavailability [12]. Previous studies have showed that the bioavailability of phenolic compounds in fruits was greatly reduced after the simulated gastrointestinal digestion, especially during intestinal digestion [13,14]. In contrast, there was research that found that simulated gastrointestinal digestion increased the bioavailability of phenolic compounds and antioxidant activity in different kinds of fruit ferment [15,16,17,18].

‘Xuehua’ pear belongs to Rosaceae pear genus (*Pyrus*), which is the main white pear variety in China [19]. It contains abundant nutrients such as carbohydrates, fat, vitamins, minerals, etc., and phenolic compounds including flavonoids, polyphenols, etc. The rich biological material base results in high free radicals scavenging [20], antioxidant activity [21], antibacterial activity [22], and other effects. ‘Xuehua’ pear is mostly eaten fresh, partly used to produce dried pear, pear wine, pear juice, etc. [23]. However, the related research of ‘Xuehua’ pear ferment has not been reported in detail. Therefore, it is necessary to study the bio-accessibility and changes of the bioactive substances, as well as the antioxidant properties in ‘Xuehua’ pear ferment during simulated gastrointestinal digestion.

In this research, the total phenolic content (TPC), total flavonoid content (TFC), protease activity, SOD activity, 1,1-diphenyl-2-trinitrophenylhydrazine (DPPH) radical scavenging activity (DPPH-RSA), hydroxyl (·OH) radical scavenging activity (·OH-RSA), and total reducing capacity were used as indicators to investigate the effect of gastrointestinal digestion in vitro on the bioactive components and antioxidant capacity of ‘Xuehua’ pear ferment. The correlation among bioactive components and antioxidant capacity were analyzed. This study aimed to provide reference for the research on the digestibility of antioxidant components of ‘Xuehua’ pear ferment.

## 2. Materials and Methods

### 2.1. Plant Materials and Strains

‘Xuehua’ pears were harvested from the commercial orchard located in Zhao County, Hebei, China. Pears with uniform maturity and free of diseases and damages were selected to produce pear ferment. *Lactobacillus plantarum* LP45-1000 was purchased from Inatural biotech (Hebei Yiran Biotechnology Co., Shanghai, China); *Saccharomyces cerevisiae* was purchased from Angel (Hubei Anqi Yeast Co., Yichang, China).

### 2.2. Standards, Reagents, and Other Chemicals

Yeast Peptone Dextrose (YPD) and Man Rogosa Sharpe (MRS) medium, analytical reagent of pepsin, trypsin, and bile extract were obtained from Yuanye Bio-Technology Co., Ltd. (Shanghai, China). Analytical reagent of citric acid, HPLC-grade of acetic acid, formic acid, and concentrated hydrochloric acid were purchased from MACKLIN (Shanghai, China). Analytical reagent of L-tyrosine, Folin-Ciocalteu’s reagent, casein, pyrogallol, glucose, sodium chloride, rutin, dibasic sodium phosphate, sodium dihydrogen phosphate, sodium bicarbonate, ethylene diamine tetraacetic acid (EDTA), Tris (hydroxymethyl) methyl aminomethane (Tris) were purchased from Sangon biotech (Shanghai, China). Phenolic and organic acid standards were purchased from Solarbio (Solarbio, Beijing, China), with the purity ≥98%. Deionized water was produced by a Milli-Q water purification system (Rephily RS2200QUV purist water system, Shanghai, China).

### 2.3. Methods

#### 2.3.1. Strain Activation and Culture

Yeast: Approximate 5 g of *Saccharomyces cerevisiae* powder was dissolved in 100 mL sterilized YPD liquid medium and shaking cultured at 28 °C for 24 h. Then, 50 μL of bacterial suspension was spread on YPD solid medium and incubated at 28 °C for 24–48 h. A single colony was picked and crossed to YPD slant medium, and stored at 4 °C for use.

*Lactobacillus plantarum*: MRS medium was used as active and cultural medium. *Lactobacillus plantarum* was cultured at 37 °C either in liquid or in solid MRS medium. Other activation and culture methods were the same as *Saccharomyces cerevisiae*.

The cryopreserved strains of *Saccharomyces cerevisiae* and *Lactobacillus plantarum* were transferred to YPD and MRS solid media, respectively, and activated at optimum temperature, then each single colony was picked and shaking cultured for 24–48 h into the corresponding liquid media, respectively. The bacteria were collected and the concentration was adjusted to 10^6^ CFU/mL by physiological saline.

#### 2.3.2. Preparation of ‘Xuehua’ Pear Ferment

The ferment preparation method was based on the results of preliminary laboratory research. Well screened ‘Xuehua’ pears were picked, washed, cored, sliced, and then mixed with equal volume (*m*/*v*) deionized water. Then 13% (*m*/*m*) glucose was added to the mixture and the pH was adjusted to 4.5 with anhydrous citric acid. The mixture was sterilized at 85 °C for 15 min and inoculated with 0.14% (*v*/*m*) 10^6^ CFU/mL *Saccharomyces cerevisiae* suspension after being cooled down to room temperature. The system was fermented for 24 h at 26 °C. The initial fermented product was inoculated with 1.5% (*v*/*m*) *Lactobacillus plantarum* suspension and incubated at 39 °C to continue fermentation for 36 h. The final product was gauze filtered, and the supernatant as ‘Xuehua’ pear ferment was stored at −20 °C for use. The pH value of ‘Xuehua’ pear ferment using pH meter S210 (METTLER TOLEDO, Shanghai, China) was 4.43, the soluble solids content, using Brix spindle PAL-1 (ATAGO, Tokyo, Japan), was 11.3 Brix%, and the total acid content using NaOH titration method (in terms of lactic acid) was 4.43 g/L. All the above methods were conformed to the general physicochemical indexes of liquid plant ferments in QB/T5323-2018 “Plant Jiaosu” standard [24].

#### 2.3.3. Simulated Gastric and Intestinal Fluid Digestion In Vitro

The method was referenced as described by Davide et al. [14] with slight modifications. The experimental design consisted of simulated gastric fluid digestion and simulated pancreas digestion. ‘Xuehua’ pear ferment and 9 mg/mL sodium chloride were mixed well by 1:4 (*v*:*v*) to make the ferment dilution. Gastric fluid digestion group (GF): 40 mL ferment dilution was adjusted pH to 2.0 ± 0.1 with 1 mol/L hydrochloric acid and added with 8 mL of simulated gastric fluid (40 mg/mL pepsin in 0.01 mol/L hydrochloric acid solution). Intestinal fluid digestion group (IF): 20 mL of gastric-digested sample was adjusted pH to 7.0 ± 0.1 with 1 mol/L sodium bicarbonate, and 4 mL of simulated intestinal fluid (0.2 g of trypsin and 1.2 g of bovine bile extract dissolved in 100 mL of 0.1 mol/L sodium bicarbonate solution, i.e., 2 mg/mL of trypsin solution and 12 mg/mL of bile extract) was added. The ‘Xuehua’ pear ferment after 2 h gastric fluid digestion was used as starting sample for simulated intestinal digestion. Samples were taken after being incubated at 37 °C for 0, 0.5, 1, and 2 h and centrifuged at 3600× *g* for 10 min. The supernatant was collected for further experiment. Digestions without enzymes and bile extract were carried out to subdivide the effects caused by the presence of enzymes or by the acid-base environment in the assay, which were defined as gastric control group (GC) and intestinal control group (IC).

#### 2.3.4. Determination of TPC

The Folin–Ciocalteu method was referenced as described by Gerardi et al. [25] with slight modifications. Samples from each sampling point were diluted 3 times by deionized water. Then, 1 mL diluted sample was added with 3.5 mL deionized water followed with 0.5 mL of Folin–Phenol reagent and 1 mL 7% (*m*/*m*) sodium carbonate. The mixture was kept at 40 °C for 30 min, and the absorbance at 765 nm was measured with deionized water as blank control. The calibration curve was plotted with gallic acid concentration (0~33.33 μg/mL) as x-axis and absorbance value as y-axis. The result was expressed as gallic acid equivalents (μg/mL).

#### 2.3.5. Determination of TFC

The aluminum nitrate colorimetric method was adapted from Chen et al. [26]. Briefly, 0.3 mL each digested sample and 0.3 mL of 5% (*m*/*m*) sodium nitrite were mixed well and let stand for 6 min in the dark. Then, 0.3 mL of 10% (*m*/*m*) aluminum nitrate was added and allowed to equilibrate for 6 min with light-free. Then, 4.0 mL of 1.0 mol/L sodium hydroxide and 0.4 mL of deionized water was mixed well with the system and stabilized for 10 min in the dark. The absorbance at 510 nm was measured with deionized water as blank control. The calibration curve was plotted with rutin concentration (16.67–83.33 μg/mL) as x-axis and the absorbance value as y-axis. The result was expressed as rutin equivalents (μg/mL).

#### 2.3.6. Protease Activity

The method was referenced as described by GB/T 23527-2009 “Protease activity determination method” (Folin method) [27]. Briefly, 1.0 mL digested sample and 1.0 mL 10 g/L casein (namely, 1.0 g of casein was dissolved in 80 mL of pH 7.5 phosphate buffer with pH adjusted by sodium hydroxide. After being in a boiling water bath for about 30 min, it was stirred until all the casein dissolved. The solution was fixed to a volume of 100 mL) and was incubated at 40 °C for 3 min, followed by reacting for 10 min at room temperature, and the reaction was stopped by adding 2.0 mL 65.4 g/L trichloroacetic acid. The mixture was kept at 40 °C for 20 min and centrifuged at 3600× *g* for 5 min, and 1.0 mL supernatant were added with 5.0 mL 42.4 g/L sodium carbonate, followed by 1.0 mL Folin–Phenol reagent. The mixed samples were kept at 40 °C for 20 min, and the absorbance was measured at 680 nm. The calibration curve was plotted with tyrosine concentration (10.00–120.00 μg/mL) as x-axis and absorbance as y-axis. The protease activity was calculated according to Equation (1).
(1)$$Protease\;activity(U/mL)=\frac{A_{1}\times 4\times n_{1}}{V\times 10}$$
where *A*_1_ is the tyrosine concentration derived from the calibration curve; 4 (mL) is the total volume of the reaction system; *n*_1_ is the dilution rate of the sample; V (mL) is the sample volume; 10 (min) is the reaction time.

#### 2.3.7. Superoxide Dismutase (SOD) Activity

The method was adapted from GB/T 5009.171-2003 “Determination of superoxide dismutase (SOD) activity in health food” [28]. Briefly, 2.35 mL A solution (pH 8.20 0.1 mol/L Tris-hydrochloric acid buffer with 1 mmol/L EDTA, pH 8.20), 1.5 mL deionized water, and 0.15 mL B solution (4.5 mmol/L pyrogallol hydrochloric acid solution) were mixed well, and the absorbance was immediately measured at the initial time and after 1 min at 325 nm, respectively. The difference in absorbance values of initial time and after 1 min was the autoxidation rate of pyrogallol, which was defined as ∆A (/min). The sample was added into the A and B solution as described above. The sample volume was adjusted to inhibit approximate half pyrogallol autoxidation rate, namely the autoxidation rate sample mixture should be about 1/2 of ∆A, and the accurate autoxidation rate of sample mixture was defined as ∆*A*_1_. The SOD activity was calculated according to Equation (2) and the value was expressed as U/mL.
(2)$$SOD\;activity\left(\frac{U}{mL}\right)=\frac{\left(∆A-∆A_{1}\right)\times 100\%\times 4.5\times D}{∆A\times 50\%\times V}$$
where ∆*A*_1_ is the autoxidation rate of sample mixture, ∆A is the autoxidation rate of blank control, V (mL) is the volume of the sample, *D* is the sample dilution times, 4.5 (mL) is the total volume of the reaction system.

#### 2.3.8. Antioxidant Capacity

##### Determination of DPPH-RSA

The method published by Romanet et al. [29] was slightly modified. Briefly, 2 mL of each digested sample and 2 mL of 0.02 mg/mL DPPH-ethanol were mixed well, and the reaction was carried out at room temperature and protected from light for 30 min. The absorbance at 517 nm was measured using ethanol as reference. DPPH-RSA was calculated according to Equation (3). The absorbance of digestion solution was used as *A_i_*, and ethanol instead of digestion solution was used as *A_c_*, and ethanol instead of DPPH-ethanol was used as *A_j_*.
(3)$$DPPH-RSA(\%)=\frac{\left[A_{c}-\left(A_{i}-A_{j}\right)\right]}{A_{c}}\times 100$$
where *A_i_* is the sample absorbance, *A_j_* is the background absorbance, and *A_c_* is the absorbance of the blank.

##### Determination of ·OH-RSA

The method published by Zacarías-Garcia et al. [30] was used. Briefly, 2 mL of each sample digestion with 2 mL of 6 mmol/L hydrogen peroxide, and 2 mL of 6 mmol/L ferrous sulfate was mixed and let stand for 10 min. Then, 2 mL of 6 mmol/L salicylic acid was added and stirred at 37 °C for 1 h. The absorbance *A*_1_ was determined at 510 nm, and the blank *A*_0_ was deionized water instead of digestion solution, and the deionized water instead of absorbance of hydrogen peroxide was *A*_2_. ·OH-RSA was calculated according to Equation (4).
(4)$$\cdot OH-RSA(\%)=\frac{\left[A_{0}-\left(A_{1}-A_{2}\right)\right]}{A_{0}}\times 100$$
where *A*_1_ is the sample absorbance, *A*_2_ is the background absorbance, *A*_0_ is the blank absorbance.

##### Total Reduction Capacity

The method was referenced as described by Liyana-Pathirana et al. [31] with slight modifications. Briefly, 1 mL of digested sample, 2.5 mL of phosphate buffer (pH 6.6), and 2.5 mL of potassium ferricyanide (1 g/100 mL) were mixed and incubated at 50 °C for 20 min. To this mixture, 2.5 mL of trichloroacetic acid 10% (*m*/*v*) was added and centrifuged at 5000× *g* for 15 min. Then, 2.5 mL of supernatant, 2.5 mL of deionized water, and 0.5 mL of ferric chloride solution 0.1% (*m*/*v*) were mixed and stabilized for 10 min. The absorbance at 700 nm was measured with deionized water as blank control, and the capacity of the total reducing power was expressed as the A_700_ value.

#### 2.3.9. Phenolic Components Analyses

The phenolic composition was analyzed using the method previously described by Su et al. [32]. HPLC analysis was performed using a HITACHI L-2000 series HPLC (Hitachi, Tokyo, Japan) equipped with a CAPCELL PAK MG C18 250 mm × 4.6 mm, 5 µm column (No. AKAD50171, OSAKA SODA, Tokyo, Japan). The samples were filtered with a 0.45 µm membrane PN4614 (Life Sciences, Hercules, CA, USA) before analysis. The analysis was maintained at a column temperature of 30 °C with an injection volume of 10 μL at a flow rate of 1.0 mL/min. The mobile phase A was methanol and mobile phase B was ultrapure water containing 0.1% (*m*/*v*) formic acid. The gradient elution was performed as follows: 0–5 min, B 95–90%, 5–30 min, B 90–80%, 30–45 min, B 80–76%, 45–60 min, B 76–61%, 60–70 min, B 61–57%, 70–80 min, B 57–53%, 80–85 min, B 53–95%, 85–5 min, B 95%. The separated phenolic components were monitored at a wavelength of 280 nm. The chromatographic peak was identified based on the retention time of the standard compound and was quantified by the peak area. The results were expressed as μg/mL.

The bio-accessibility [33] of phenolic components under simulated gastrointestinal digestion was determined using the following Equation (5):(5)$$Bio-accessibility(\%)=\frac{Finalconcentration]}{Finalconcentration}\times 100$$
where the final concentration is the phenolic components content at the end of the gastric and intestinal phases, the initial concentration is the phenolic components content in the undigested pear ferment.

#### 2.3.10. Organic Acids Analyses

Individual organic acids were quantified as described by Wojdylo et al. [34] using a HITACHI L-2000 series HPLC (Hitachi, Tokyo, Japan) equipped with a ADME HR C18 250 mm × 4.6 mm, 5 µm column (Shiseido, Tokyo, Japan). Prior to injection into the HPLC, samples were filtered through a 0.45 µm supor membrane PN4614 (Life Sciences, Hercules, California, USA). The mobile phase was ultrapure water containing 0.23% (*m*/*v*) Ammonium dihydrogen phosphate. UV detection at a wavelength of 210 nm and a flow rate of 0.8 mL/min were used for the analysis of organic acids. Data were analyzed using the HSIII Hitachi L-2000 software (Hitachi, Tokyo, Japan). Identification was made by comparing retention times of commercial pure standards, and quantification was based on the UV signal response. Organic acid content was expressed in mg/mL.

### 2.4. Statistical Analysis

Excel 2021 (Microsoft, Redmond, USA) and SPSS 21.0 (IBM, Chicago, IL, USA) were used to analyze the data. Pearson’s method [35] for correlation analysis was performed by the Genescloud platform (Personalbio, Shanghai, China). Data were presented as means ± standard deviations (SD) of at least three independent experiments; each experiment had a minimum of three replicates of each sample.

## 3. Results

### 3.1. Effect of Simulated Digestion on TPC In Vitro

The calibration curve of TPC determination was obtained as y = 0.0812x + 0.1082 (R^2^ = 0.999, linear range with 0–33.33 μg/mL), where x was gallic acid concentration and y was the absorbance value. The effects of different digestion treatments on the TPC of ‘Xuehua’ pear ferment are shown in Figure 1. The TPC of pear ferment decreased after gastric and intestinal digestion, in comparison to the initial sample of pear ferment. During the gastric digestion, the TPC of the GF was stable within 1 h of digestion, and then increased significantly (*p* < 0.05), reaching a maximum of 18.7 μg/mL at 2 h of gastric digestion. There was an increase when the sample transformed into the simulated intestinal environment, but the whole intestinal digestion phase showed a decreasing trend. The TPC of the IF decreased significantly (*p* < 0.05) from 0 to 0.5 h, and then stabilized, with the content of 17.9 μg/mL at 2 h. The TPC of the GC decreased significantly with the digestion time (*p* < 0.05), while in IC it decreased significantly within 1 h of digestion (*p* < 0.05), but it did not change significantly thereafter.

### 3.2. Effect of Simulated Digestion on TFC In Vitro

The calibration curve of TFC determination was obtained as y = 4.923x + 0.1133 (R^2^ = 0.9993, linear range with 16.67–83.33 μg/mL), where x was rutin concentration and y was the absorbance value. The results reported in Figure 2 show the effects of different digestion treatments on the TFC of ‘Xuehua’ pear ferment. The TFC of the pear ferment increased significantly after gastrointestinal digestion (*p* < 0.05). There was a time-dependent increase of the TFC in GF (*p* < 0.05), and the content reached the maximum value of 198.3 μg/mL at 2 h of digestion. There was no significant change in TFC after the transfer of the sample from gastric phase to intestinal phase. The TFC remained stable within 0.5 h of digestion in IF, after which it decreased significantly (*p* < 0.05), with a minimum of 161.7 μg/mL at 2 h of digestion. The trends of the TFC of the GC and IC were consistent with the GF and IF, but their contents were lower than that in GF and IF.

### 3.3. Effect of Simulated Digestion on Protease Activity In Vitro

The calibration curve of protease activity determination was obtained as y = 0.0105x − 0.0471 (R^2^ = 0.9982, linear range with 10.00–120.00 μg/mL), where x was tyrosine concentration and y was the absorbance value. The changes of protease activity of ‘Xuehua’ pear ferment are shown in Figure 3. The protease activity of the initial pear ferment sample was significantly increased after gastrointestinal digestion. The protease activity of the GF increased significantly (*p* < 0.05) within 0.5 h of gastric digestion, and the value was 114 U/mL at 0.5 h, while nothing changed significantly thereafter. There was a decline when transferred into the intestinal fluid, but the whole intestinal digestion phase was on the rise. The protease activity increased significantly in IF within 0.5 h of digestion (*p* < 0.05), and then kept constant until the end of the gastric phase, with the final protease activity of 161 U/mL. The protease activity of the GC and IC did not change significantly throughout the digestion phase.

### 3.4. Effect of Simulated Digestion on SOD Activity In Vitro

The SOD activity increased during gastric digestion and decreased during intestinal digestion. During gastric digestion, SOD activity in GF increased at first and then decreased (*p* < 0.05), with the highest activity of 14,400 U/mL at 1 h of digestion. There was a significant decline when transferred into the intestinal environment (*p* < 0.05), with the SOD activity of 4800 U/mL at initial intestinal digestive phase. The SOD activity in IF decreased significantly with digestion time (*p* < 0.05), and the lowest activity was 1600 U/mL at 2 h of digestion in IF. The trend of the SOD activity of the GC and IC was consistent with that in GF and IF, but the activity in control group was lower (Figure 4).

### 3.5. Effect of Simulated Digestion on Antioxidant Capacity In Vitro

#### 3.5.1. Changes in DPPH-RSA during Simulated Digestion In Vitro

The effects of different digestion treatments on DPPH-RSA of ‘Xuehua’ pear ferment are shown in Figure 5. The DPPH-RSA increased after gastric digestion and decreased in intestinal digestion. During gastric digestion, the DPPH-RSA in GF increased significantly within 1 h (*p* < 0.05) and reached the highest as 99.75% at 1 h of digestion. At 2 h of digestion, the DPPH-RSA decreased (*p* < 0.05) but was not significantly different from that of 0 h of GF digestion. There was an obvious reduction when the digestive environment altered, with the DPPH-RSA of 40.25%. The DPPH-RSA in IF decreased significantly within 0.5 h (*p* < 0.05), and activity remained constant thereafter, with a scavenging rate of 36.71% at 2 h of digestion. The DPPH-RSA of the GC showed a similar trend with GF with lower content. The DPPH-RSA in IC decreased significantly with the digestion time (*p* < 0.05).

#### 3.5.2. Changes in ·OH-RSA during Simulated Digestion In Vitro

Figure 6 shows the changes in ·OH-RSA of ‘Xuehua’ pear ferment during in vitro gastrointestinal digestion. The ·OH-RSA decreased after gastric digestion while it increased significantly after intestinal digestion. During gastric digestion, the ·OH-RSA in GF gradually increased with the digestion time (*p* < 0.05), reaching the highest as 33.24% at 2 h of digestion. It increased to 97.44% when transferred into the intestinal phase of digestion, while the whole intestinal digestion phase had a decreasing trend, and the final ·OH-RSA was 94.69%. There was a time-dependent increase of the ·OH-RSA in GC within 1 h of digestion (*p* < 0.05), after that, it was significantly lower (*p* < 0.05). The trend of the ·OH-RSA of the IC was consistent with the IF.

#### 3.5.3. Changes in the Capacity of Total Reducing during Simulated Digestion In Vitro

As shown in Figure 7, the total reducing capacity decreased after gastrointestinal digestion compared with the undigested sample. During the gastric digestion, the total reducing capacity of the GF gradually increased with the digestion time (*p* < 0.05), reaching the maximum as 0.45 at 2 h of digestion, which was 62.92% of the undigested sample. It decreased to 0.34 when the sample was transferred to the simulated intestinal environment. The total reducing capacity in IF gradually decreased with the digestion time (*p* < 0.05). At 2 h of digestion, the total reducing capacity decreased to 0.32. The change trends of the total reducing capacity of the GC and IC were consistent with the GF and IF.

#### 3.5.4. Correlation Analysis between Bioactive Ingredients and Antioxidant Activity

Numerous studies have shown that the antioxidant capacity of plants is related to their active ingredients such as total phenolics and flavonoids [36,37]. Correlation analyses of TPC, TFC, protease and SOD activity, DPPH-RSA, ·OH-RSA, and total reducing capacity of ‘Xuehua’ pear ferment were performed to investigate the interactions among them during the simulated gastrointestinal digestion.

According to the correlation analysis (Figure 8), it was found that TPC, TFC, and SOD activity in the ferment simulated gastro-intestinal digestion showed significant positive correlation with the ·OH-RSA and total reducing capacity (*p* < 0.05). In the gastric digestion phase, there were two clusters that showed highly positive correlations respectively, which were the SOD activity, protease activity, TFC cluster and the TPC, ·OH-RSA, total reducing capacity cluster. However, the DPPH-RSA showed a highly significant negative correlation with the TPC (*p* < 0.01), and negatively correlated with other items. In the simulated intestinal digestion phase, protease activity was significantly negatively correlated with DPPH-RSA, ·OH-RSA, and total reducing capacity (*p* < 0.01), and the positive-related clusters were the TPC, DPPH-RSA cluster, and the TFC, SOD activity, ·OH-RSA-total reducing capacity cluster.

### 3.6. Effect of Simulated Digestion on Phenolic Components In Vitro

Phenolic compounds considered to be biologically effective are released from the food matrix through enzymatic action in the digestive fluids [12]. In order to investigate and clarify the changes in the individual phenolic components of ‘Xuehua’ pear ferment during gastrointestinal digestion, a total of seven components were evaluated by HPLC: protocatechuic acid, chlorogenic acid (phenolic acid and its derivatives), arbutin, epicatechin, hyperoside, isorhamnetin-*3-O*-glucoside, and quercetin (flavonoids and their derivatives). The results of the quantitative analysis of these phenolic components during gastrointestinal digestion are shown in Figure 9.

The phenolic components in pear ferment were mainly arbutin and chlorogenic acid. The content of arbutin and chlorogenic acid increased from 66.13, 36.58 μg/mL at initial to 68.21, 37.62 μg/mL at 2 h of gastric digestion, respectively. During the in vitro-simulated gastric digestion, the content of the epicatechin was gradually increased with the digestion time (*p* < 0.05) and reached the highest content (0.74 μg/mL) at 1 h of digestion but became undetectable at 2 h of digestion. The content of hyperoside and isorhamnetin-*3-O*-glucoside decreased from the initial maximum value 1.46 and 3.65 μg/mL at initial to 0.41 and 3.25 μg/mL, after 2 h of digestion, respectively. The content of protocatechuic acid showed an increasing trend with the whole gastric digestion phase (*p* < 0.05), with the maximum value of 6.50 μg/mL at 2 h of digestion.

When transferred to the intestinal environment, the content of phenolic components was significantly decreased in comparison to the final content after GF, and additionally the new substance quercetin was detected. During the in vitro-simulated intestinal digestion, both arbutin and quercetin increased in content with increasing digestion time, with the highest content at 2 h of digestion, 40.60 and 1.81 μg/mL, respectively. The content of protocatechuic acid first increased and then decreased, at the highest content of 0.25 μg/mL at 1 h of digestion and was not detected after 2 h of digestion. While the content of hyperoside was 0.36 μg/mL at 0 h and was not detected thereafter. The contents of chlorogenic acid and isorhamnetin-*3-O*-glucoside decreased, with chlorogenic acid at a minimum of 0.19 μg/mL at 0.5 h of digestion, after which it was undetectable. The content of isorhamnetin-*3-O*-glucoside was 0.70 μg/mL at 0 h of digestion, after which it was undetectable.

Figure 9c shows the bio-accessibility of the phenolic components after the simulat-ed gastric and intestinal digestion. The bio-accessibility of arbutin, protocatechuic acid, chlorogenic acid, and isorhamnetin-*3-O*-glucoside increased after gastric digestion, while hyperoside decreased. Epicatechin was not detected at the end of gastric digestion and the whole intestinal phase, thus its bio-accessibility was 0. After intestinal digestion, arbutin decreased to 61.38% and other phenolic components were not detectable, except for quercetin. Quercetin was not detected in the original pear ferment and gastric phase, but it increased during intestinal digestion, therefore its bio-accessibility could not be calculated.

### 3.7. Effect of Simulated Digestion on Organic Acids In Vitro

Studies have shown that organic acids in plants have a direct effect on scavenging of free radicals and other antioxidant capacities [34]. To further investigate and clarify the changes of organic acids in ‘Xuehua’ pear ferment during the simulated gastrointestinal digestion, oxalic acid, tartaric acid, quinic acid, malic acid, lactic acid, acetic acid, and citric acid as the major organic acids in ‘Xuehua’ pear ferment were quantified (Figure 10).

The organic acids in pear ferment were mainly lactic acid and citric acid, with contents of 4.96 and 1.77 μg/mL, respectively. When transferred to the simulated gastric environment, the lactic acid and citric acid contents all increased. During the in vitro-simulated gastric digestion, both malic and acetic acid contents showed an increasing trend, with no significant change in malic acid content within 0.5 h (*p* > 0.05) and the highest content of 0.64 mg/mL at 2 h of digestion (*p* < 0.05), and the acetic acid content stabilized after 1 h of digestion and reached 0.17 mg/mL at 2 h. The content of quinic acid and citric acid decreased from 2.24 and 3.31 mg/mL at 0 h to 0.85 and 2.94 mg/mL after 2 h of digestion, respectively. The content of the oxalic, tartaric, and lactic acid did not vary significantly throughout the digestion phase, with the highest levels of 0.18, 1.75, and 6.55 mg/mL at 2 h of digestion.

When transferred to the simulated intestinal environment, the content of organic acids decreased, in comparison to their final content after the gastric phase, except for malic acid. During the in vitro-simulated intestinal digestion, there was a time-dependent increase of the contents of tartaric acid and acetic acid (*p* < 0.05), and the highest contents were 2.06 and 0.10 mg/mL at 2 h of digestion, respectively. The oxalic acid and lactic acid contents showed an increasing trend and stabilized after 0.5 h of digestion, with the highest contents of 0.15 and 3.63 mg/mL at 2 h of digestion. The quinic acid content did not change significantly within 1 h of digestion (*p* > 0.05), after which it decreased significantly to 0.47 mg/mL at 2 h of digestion. The malic acid content gradually decreased with digestion time, and the content decreased to 0.12 mg/mL at 2 h of digestion. The citric acid content did not change significantly throughout the digestion period and was 2.66 mg/mL at 2 h of digestion.

## 4. Discussion

Current studies on human nutritional intake have mainly focused on the extraction and function of bioactive ingredients in foods, without considering release and possible changes of the bioactive ingredients during gastrointestinal digestion. For example, Bouayd et al. [38] reported the TPC and TFC after simulated gastrointestinal digestion in apples were only 45% and 60% of the total methanol extract. It indicates that the bio-accessibility of bioactive ingredients in the food matrix are affected by their release, stability, and the interactions among substances. In this study, TPC, TFC, protease activity, SOD activity, and antioxidant capacity in ‘Xuehua’ pear ferment during in vitro-simulated gastrointestinal digestion were investigated to provide a theoretical basis for evaluating the bio-accessibility of ‘Xuehua’ pear ferment.

Total phenolics contain all phenolic substances, including mono- and polyphenols. Pear fruit is rich in phenolics, especially flavonoids (have a common C_6_-C_3_-C_6_ skeleton) and phenolic acids [39]. Phenolic substances have various physiological effects such as antioxidant, anti-aging, as well as reducing many diseases including cancer, cardiovascular disease, diabetes, and chronic respiratory diseases [40,41]. The TPC of pear ferment decreased after gastric and intestinal digestion. According to the study, the TPC in green tea was significantly reduced in the initial phase of gastric digestion by a low-acid environment, and it was suggested that the reduction in TPC may be related to the instability of macromolecular phenolic components upon entering a low-acid environment, suggesting that pH plays an important role in altering the content of bioactive compounds [42]. The TPC showed an increasing trend through gastric digestion, combined with the increased content of protocatechuic acid, chlorogenic acid, and arbutin in pear ferment (Figure 9a). The GC group showed a decreasing trend over time, but the decrease was lower than that of pear ferment. This was possibly because the polymerization of polyphenols was alleviated at low pH; meanwhile, the addition of pepsin allowed most of the combined phenolic components to be released and detected [43]. The result was consistent with previous studies [38,44,45]. The TPC carried over the increase when the sample was transferred into intestinal digestion, but it rapidly decreased after 0.5 h. Chen et al. [46] found a similar result and assumed that the phenolic components contents were decreased or conversed to other substances when the environment changed from acidic to mild alkaline intestinal. Ryan et al. [47] suggested that this could be due to structural transformation of polyphenols. The pear ferment was fermented by yeast and *Lactobacillus plantarum*, which consumes carbohydrates, reducing sugars and other nutritional substrates, and produces lactic acid, making the overall environment moderately acidic. The racemization of molecular structure increased with pH, producing two chiral enantiomers with different reactions, while the intestinal environment was weakly alkaline, and the structural change could result in polyphenols being undetectable [47,48]. The present result, that the decrease and the disappearance of chlorogenic acid and isorhamnetin-*3-O*-glucoside during intestinal digestion, as well as the conversion of hyperoside to quercetin (Figure 9b), was in agreement with the result of Jara-Palacios et al. [49] in white wine extracts. The IC group showed a sustained decrease throughout the intestinal digestion. It was considered that trypsin and bile extracts allowed the formation of certain interactions between phenolic components monomers, which increased their stability in the intestinal fluid and led to the stabilization of the TPC [50].

The TFC increased in GF (*p* < 0.05), and GC showed a similar trend. This indicated that the acidic environment at low pH was more favorable to reduce the intermolecular force for the decomposition of the flavonoids in the ferment, as arbutin (Figure 9a) [14]. The TFC remained stable within 0.5 h of digestion in IF (*p* > 0.05), and decreased significantly thereafter (*p* < 0.05), which was in conformity with IC. Epicatechin was not detected in IF (Figure 9b). One of the possible reasons was that the flavonoids catechin and epicatechin, which were rich in pears, were unstable at high pH [36]. Tenore et al. [51] found a loss of content in 91.8% of catechins during simulated intestinal digestion, suggesting that the alkaline environment and dissolved oxygen in the intestine might cause the dimeric autoxidation of catechins.

Protease and SOD are common potent enzymes in pear ferment. Protease catalyzes the hydrolysis of proteins to produce peptides and small molecule amino acids [52,53]. SOD is one of the important antioxidant enzymes [54], which can scavenge reactive oxygen species from the body and improve immunity [55]. Both are often used as important indicators for judging the nutritional value of ferments. The protease activity of undigested ferment increased during gastric digestion, and the GF increased significantly especially for the first 0.5 h (*p* < 0.05). That was related to the pepsin-assisted protease hydrolysis of the protein complex class in the ferment, which in turn sped up the enzymatic reaction until stabilization [56]. However, since the significant increase in pear juice matrix and fermented lactic and acetic acid content provided an acidic environment, it is presumed that the bioactive enzymes produced by pear ferment were suitable for low pH. There was an obvious decline in both the IF and IC, indicating that the protease produced by the pear ferment could be lost or unstable when subjected to pH changes [57], while there is a rapid rise subsequently observed in the digestion with enzyme and bile extract. Studies have shown that phenolic compounds could inhibit various enzymes [58], thus the decrease of TPC might lead to the rise of protease activity. In addition, studies also found that intestinal environment could affect protease activity so that the digestive enzyme and the extracellular enzyme produced by probiotic bacteria could further break down the proteins in the matrix to enhance the protease activity [59].

The SOD activity initially increased and finally decreased in gastric digestion, while in GC it was not obvious. This indicates that SOD activity was inhibited by the acidic environment [60], but the addition of pepsin changed the digestive substrate environment, prompting microbial growth and metabolism to secrete the corresponding efficacy enzymes [61]. There was a significant decline when the sample was transferred into the simulated intestinal environment and decreased significantly in IF with digestion time. It is presumed that some reactive groups in the ferment co-hydrolyzed by trypsin and bile extracts made a competitive inhibition to SOD, resulting in a significant decrease in SOD activity [56]. The change trend in SOD activity is consistent with that in lychee juice during the simulated intestinal digestion in vitro [56].

DPPH radical is a synthetic and stable organic nitrogen radical with high sensitivity [62], while ·OH radical is a highly toxic radical with the most active chemical properties, especially in the presence of metal ions, which can damage proteins, nucleic acids, and other macromolecular nutrients [43]. DPPH-RSA and ·OH-RSA are commonly used to evaluate antioxidant capacity. During gastric digestion, the DPPH-RSA in GF increased significantly (*p* < 0.05), which was not observed in GC. This indicated that the acidic environment had less effect on the DPPH-RSA in the samples while pepsin was the main factor. The DPPH-RSA finally decreased at 2 h of digestion, which was in agreement with the findings of Xia et al. [63] and Su et al. [64]. There was an obvious reduction when the digestive environment altered to the intestine and the IF decreased significantly within 0.5 h. Jara-Palacios et al. [49] determined the DPPH-RSA of white wine extracts after gastrointestinal digestion and found it much lower than that of the undigested samples, which is consistent with the results of this study. The changes from gastric to intestinal digestion might be related to the number and location of the hydroxyl groups of flavonoids in the ferment. It was found that trypsin could reduce the scavenging capacity of citrus flavonoids for DPPH-RSA by masking some of the hydroxyl groups in citrus flavonoids (naringin, neohesperidin, and hesperidin) [65]. In addition, it has been shown that the content reduction caused by the interaction of phenolic acids in the gastrointestinal tract also affects the DPPH-RSA [66]. When the digestion process proceeded, phenolics were released by hydrolysis, and antioxidant active substances such as phenolic hydroxyl groups increased sequentially [67]. The overall performance of DPPH-RSA was showed to be highly consistent with SOD activity, which indicated that SOD could be an important factor to affect DPPH-RSA during the gastrointestinal digestion in pear ferment [68].

The ·OH-RSA was decreased when brought into gastric digestion, and this was highly consistent with TPC. Afterwards, there was a time-dependent increase of the ·OH-RSA in GF as well as in GC, which indicated that both pepsin and acidic conditions could affect the ·OH-RSA of the ferment. The hydrolysis of bound polyphenols under low pH and digestive enzymes might be the main reason to improve the ·OH-RSA during gastric digestion [68]. The average of the ·OH-RSA in IF was much higher than that in GF, while the whole intestinal digestion phase was on the decrease. Davide et al. [14] found that gastrointestinal digestion could promote the release of antioxidant active substances in grapes to enhance their antioxidant activity, and the same findings were reported by Pavan et al. [69] and Huang et al. [70]. The decrease trend during the whole intestinal digestion was highly correlated TPC, TFC, and SOD activity. The same conclusion was found in simulated in vitro-intestinal digestion of lychee juice [56]. DPPH radical is a nitrogen radical, while hydroxyl radical is an oxygen radical. The reaction mechanism for scavenging different radicals may involve the transport of electrons or hydrogen atoms. Therefore, the ability of the same substance to scavenge these two radicals was different. It was also found that the high DPPH-RSA existed in simulated gastric digestion, and the high ·OH-RSA in simulated intestinal digestion of mango ferment [71].

The determination of reducing power is commonly done to evaluate the activity of antioxidants, by testing whether the sample has a good electron supply. Bioactive materials can scavenge free radicals in the body by giving electrons to act on chain reactions, and the reducing power as well as the antioxidant capacity are decided by the electron supplying power [54]. The total reducing capacity decreased when carried into gastric digestion compared with the undigested sample, and gradually increased with digestion time in GF, which was consistent with TPC and ·OH-RSA. It decreased when the environment changed to intestinal digestion. This was highly consistent with the variation pattern of TPC, TFC, and SOD activity, indicating that these components might be the important factors for the reducing function, which was supported by the findings of Davide et al. [14].

However, the DPPH-RSA showed a highly significant negative correlation with the TPC (*p* < 0.01) and was negatively correlated with other items during gastric digestion. Wang et al. [44] found the negative correlation between DPPH-RSA and most polyphenols in apple juice, except for epicatechin, and suggested that chlorogenic acid could reduce the correlation between phenolics and DPPH-RSA. Epicatechin is a member of procyanidins and has been reported to be a strong reducing reagent and highly correlated with DPPH-RSA [72,73]. It was also confirmed in this study that epicatechin and DPPH-RSA showed the consistent trend in GF. In the simulated intestinal digestion phase, protease activity was significantly negatively correlated with TPC, DPPH-RSA, ·OH-RSA, and total reducing capacity (*p* < 0.01). Studies have shown that certain proteins in food could promote the bioavailability of phenolic substances, and improve the antioxidant activity [74,75]. Protease could affect the available protein content and lead to the decrease of TPC and antioxidant capacity. In conclusion, the total phenolics, total flavonoids, protease and SOD activity of ‘Xuehua’ pear ferment contributed to its antioxidant capacity, while the antioxidant capacity of the ferment also varies depending on the fermentation substrate and digestive environment.

The most abundant phenolic components and organic acids in the pear ferment were arbutin, chlorogenic acid, lactic acid, and citric acid, and these metabolites have been reported to be potent antioxidant, antibacterial, anti-inflammatory, and anticancer potential agents. [76,77,78]. After gastric digestion, their average levels all increased, and these components were degraded during intestinal digestion, mainly due to the pH conditions of each digestion stage, which could hinder their biological activity [13,14], while the average levels of citric acid were still higher than those of pear ferment. The phenolic components and organic acids such as arbutin, protocatechuic acid, chlorogenic acid, tartaric acid, malic acid, lactic acid, and acetic acid were released during simulated gastric digestion, and enhanced TPC and TFC and antioxidant capacity of pear ferment, which is similar to the results of Jiao et al. [13] and Davide et al. [14]. While the content of epicatechin was low, moreover, it was not quantifiable after 2 h of gastric digestion with poor stability. Tenore et al. [51] suggested that epicatechin was affected by the dimeric autotrophy under the acidic pH of the gastrointestinal tract, which was consistent with this study. In addition, some phenolic components and organic acids were also susceptible in acidic or enzymatic reactions, as evidenced by the decrease of hyperoside, isorhamnetin-*3-O*-glucoside, quinic acid, and citric acid, which was in agreement with the findings of Engin et al. [45] and Bouayed et al. [38]. The shift of the environment from acidic gastric fluid to mild alkaline intestinal fluid led to a decrease in bioavailability, performing in the decrease of arbutin, protocatechuic acid, chlorogenic acid, hyperoside, isorhamnetin-*3-O*-glucoside, oxalic acid, tartaric acid, quinic acid, lactic acid, acetic acid, and citric acid, which is consistent with most studies [13,14]. This is mainly because the glycosidic ligand backbones were susceptible to be degraded into different structural and chemical forms under alkaline conditions [45]. For example, quercetin could be obtained from hyperoside by removing the glycosidic bond [79], resulting from the decrease in glycosidic and the increase in quercetin content throughout the intestinal digestion phase. However, there was a rising trend of digestion of arbutin, quercetin, oxalic acid, tartaric acid, lactic acid, and acetic acid during the simulated intestinal digestion. It has been shown that the addition of trypsin and bile extract mostly act on the combined phenolics, and some neutral or basic phenolics could be stabilized or released in the intestinal fluid medium [80]. Organic acids are small molecule hydrophilic compounds, and their release is facilitated by the large number of microorganisms and related enzymes in the digestive environment [81].

## 5. Conclusions

Fruit ferment is an emerging functional food, and studies about the effects of simulated digestion on the full bioactive substances and enzyme activities of pear ferment are scarce. In this study, we investigated the effect of in vitro-simulated gastrointestinal digestion on the bioactive components and antioxidant capacity of ‘Xuehua’ pear ferment. TPC, TFC, protease activity, ·OH-RSA, and total reducing capacity showed an increasing trend during the simulated gastric digestion, whereas the SOD activity and DPPH-RSA and epicatechin content showed first an increasing and then a decreasing trend, and hyperoside, isorhamnetin-*3-O*-glucoside, quinic acid, and citric acid decreased (*p* < 0.05). The transformation of the intestinal environment led to a decrease of polyphenols in most analyzed categories, except for protease activity, quercetin, and tartaric acid, while the decrease stabilized in TPC and DPPH-RSA due to the addition of bile extract and digestive enzymes. The bioactive components contributed to the antioxidant ability of the pear ferment, while the ferment matrices and digestive environment were also affected. The changes of bioactive substances during the simulated gastrointestinal digestion further indicated that the gastrointestinal tract could also be an active ingredient extractor, as the acid-base environment and the digestive enzymes contributed to the extraction of bioactive ingredients such as phenolic components from the ferment. Therefore, ‘Xuehua’ pear ferment could be considered as a kind of digestive functional food. Furthermore, whether differences in the stability of various bioactive ingredients in gastrointestinal digestion were related to their own nature, degradation pathways, metabolites, or the interaction among the active ingredients deserves further investigation.

## Figures and Tables

**Figure 1 foods-12-01211-f001:**
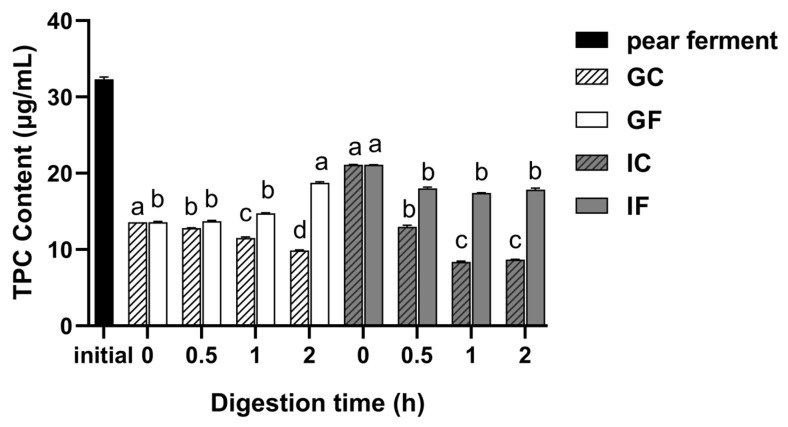
Changes of TPC of ‘Xuehua’ pear ferment during the simulated gastric and intestinal digestion in vitro. Different letters indicate significant differences in the same group under different digestion time (*p* < 0.05). Data shown as means ± SD of three replicates (*n* = 3).

**Figure 2 foods-12-01211-f002:**
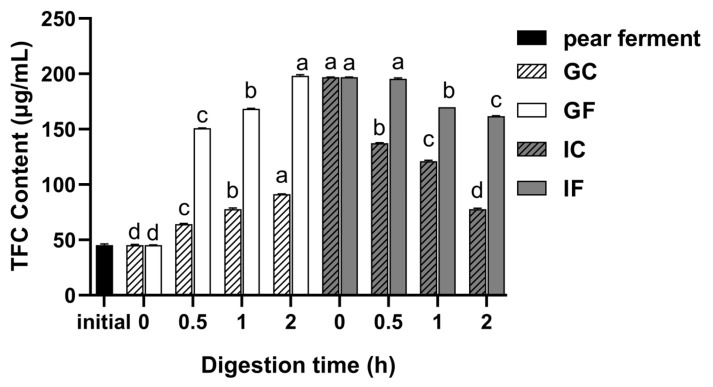
Changes of TFC of ‘Xuehua’ pear ferment during the simulated gastric and intestinal digestion in vitro. Different letters indicate significant differences in the same group under different digestion times (*p* < 0.05). Data shown as means ± SD of three replicates (*n* = 3).

**Figure 3 foods-12-01211-f003:**
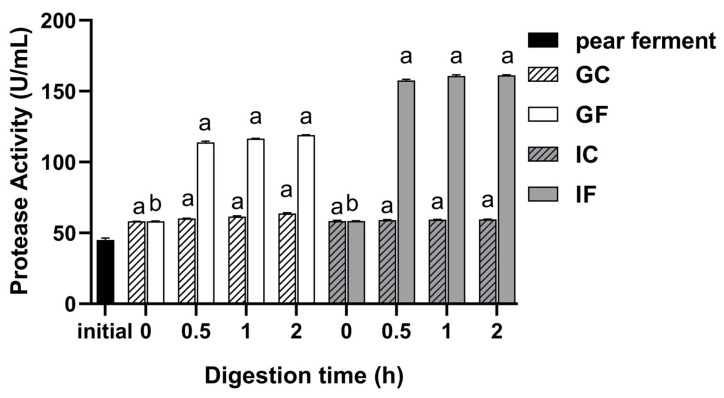
Changes of protease activity of ‘Xuehua’ pear ferment during the simulated gastric and intestinal digestion in vitro. Different letters indicate significant differences in the same group under different digestion times (*p* < 0.05). Data shown as means ± SD of three replicates (*n* = 3).

**Figure 4 foods-12-01211-f004:**
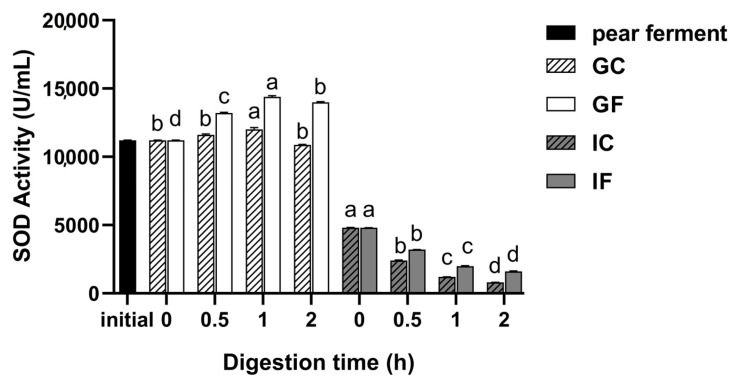
Changes of SOD activity of ‘Xuehua’ pear ferment during the simulated gastric and intestinal digestion in vitro. Different letters indicate significant differences in the same group under different digestion times (*p* < 0.05). Data shown as means ± SD of three replicates (*n* = 3).

**Figure 5 foods-12-01211-f005:**
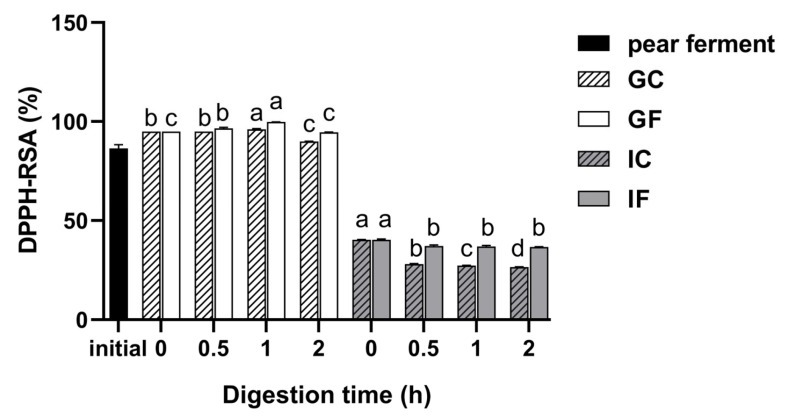
Changes of DPPH-RSA of ‘Xuehua’ pear ferment during the simulated gastric and intestinal digestion in vitro. Different letters indicate significant differences in the same group under different digestion times (*p* < 0.05). Data shown as means ± SD of three replicates (*n* = 3).

**Figure 6 foods-12-01211-f006:**
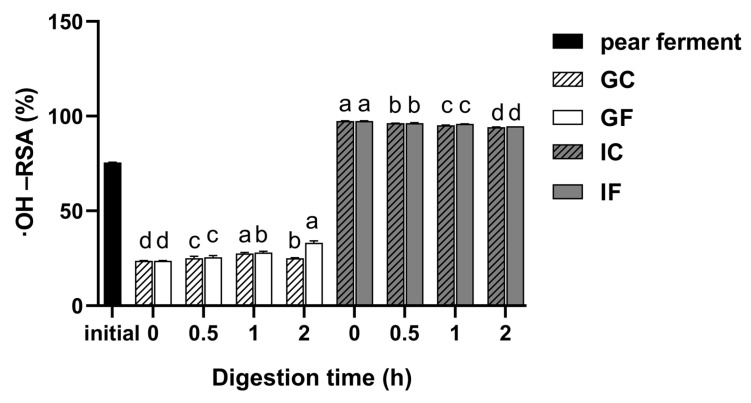
Changes of ·OH-RSA of ‘Xuehua’ pear ferment during the simulated gastric and intestinal digestion in vitro. Different letters indicate significant differences in the same group under different digestion times (*p* < 0.05). Data shown as means ± SD of three replicates (*n* = 3).

**Figure 7 foods-12-01211-f007:**
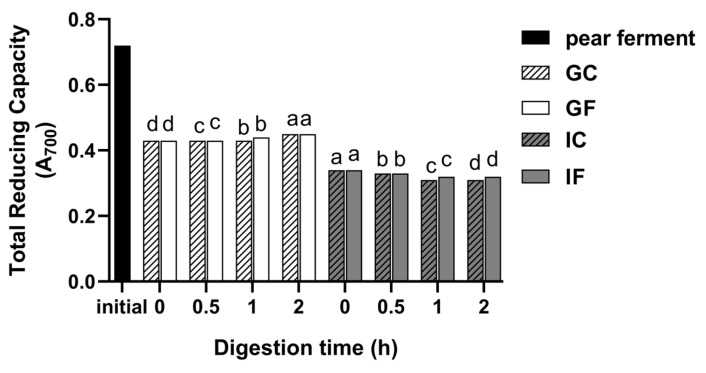
Changes of total reducing capacity of ‘Xuehua’ pear ferment during the simulated gastric and intestinal digestion in vitro. Different letters indicate significant differences in the same group under different digestion times (*p* < 0.05). Data shown as means ± SD of three replicates (*n* = 3).

**Figure 8 foods-12-01211-f008:**
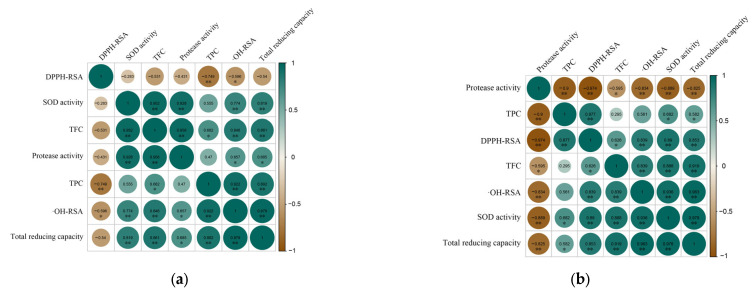
Correlation analysis between the active components of ‘Xuehua’ pear ferment and antioxidant capacity during the simulated digestion in vitro. (**a**) indicates simulated gastric digestion; (**b**) indicates simulated intestinal digestion. * indicates significant correlation (*p* < 0.05), ** indicates significant correlation (*p* < 0.01).

**Figure 9 foods-12-01211-f009:**
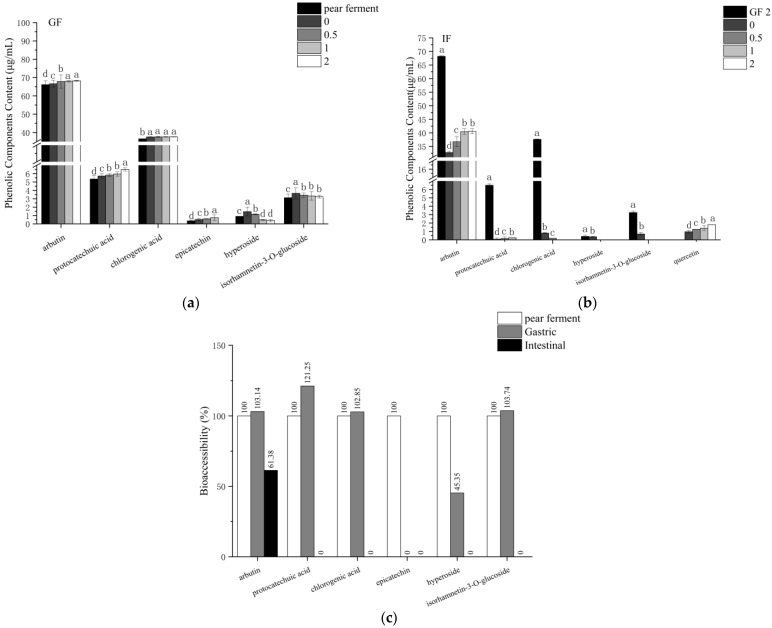
Changes of phenolic components of ‘Xuehua’ pear ferment during the simulated digestion in vitro. (**a**) indicates simulated gastric digestion; (**b**) indicates simulated intestinal digestion; (**c**) indicates bio-accessibility subjected to simulated in vitro digestion. Different letters indicate significant differences in the same group under different digestion times (*p* < 0.05). Data shown as means ± SD of three replicates (*n* = 3).

**Figure 10 foods-12-01211-f010:**
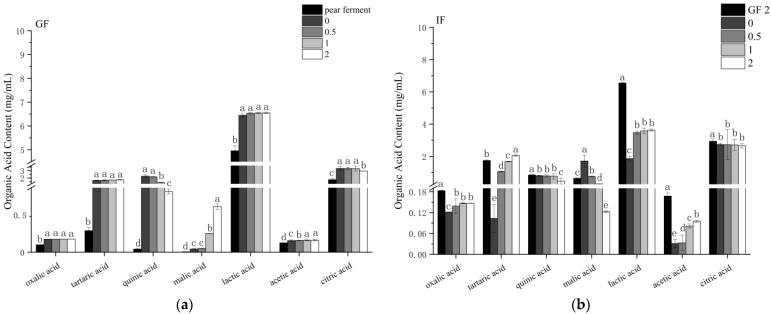
Changes of organic acids of ‘Xuehua’ pear ferment during the simulated digestion in vitro. (**a**) indicates simulated gastric digestion; (**b**) indicates simulated intestinal digestion. Different letters indicate significant differences in the same group under different digestion times (*p* < 0.05). Data shown as means ± SD of three replicates (*n* = 3).

## Data Availability

The data presented in this study are available on request from the corresponding author.

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
