# Peer review of "Changes of Bioactive Components and Antioxidant Capacity of Pear Ferment in Simulated Gastrointestinal Digestion In Vitro"

_foods, 2023, doi:10.3390/foods12061211_

Round 1
Reviewer 1 Report
Comments:
1) list all chemicals used, including their purity, and suppliers in chapter 2.2
2) the results of the calibration analyzes should not be entered in the Materials and methods chapter, but in the Results and discussion
3) unclearly described calculation of protease activity (equation 1)
4) in general, the description of the methods is not very clear (see notes in text)
5) rewrite the chapters dealing with HPLC analysis (2.3.9 and 2.3.10), they are incomprehensible (consult a chromatography expert).
6) Please explain the difference in the TPC content in the pear ferment and the sample at zero time for GF and/or IF (Figure 1)?
7) Present the results (TPC, TFC, SOD, protease activity, etc.) to the number of valid places corresponding to the precission of method! Presentation of results of TPC or TFC at hundredths of mg/mL is nonsense with respect to LOD.
8) The content of TPC in the GF2 sample (expressed as gallic acid) is lower (ca. 18 mg/mL, Fig. 1) than the content of gallic acid in the same sample determined by the HPLC method (ca. 75 mg/mL, Fig. 9b). Please explain.
9) Is the content of acetic acid in the range of 130-170 mg/mL realistic (Fig. 10)? Aren't the wrong units listed as milligrams instead of micrograms?
Author Response
Dear Reviewer,
Thank you for your valuable comments.
We carefully read the comments of all reviewers and revised the article as follows:
- Some parts of method, result and discussion were rewritten to make the description more clear and accurate.
- The data of digestive control group was added as required. Some inaccurate data was corrected.
- We enhanced English expression and modified some confusing sentences.
We put the manuscript in word with ‘track change’ function to show where we revised. Please see the attachment.
Here I would like to answer your questions and suggestions on this paper.
1) list all chemicals used, including their purity, and suppliers in chapter 2.2
Response 1: All the chemical reagent information had been added as required.
2) the results of the calibration analyzes should not be entered in the Materials and methods chapter, but in the Results and discussion
Response 2: Calibration curves had been moved to the Result part, and linear range had been added.
3) unclearly described calculation of protease activity (equation 1)
Response 3: We had revised description of protease activity.
4) in general, the description of the methods is not very clear (see notes in text)
Response 4: We had revised the methods as suggested in the text.
5) rewrite the chapters dealing with HPLC analysis (2.3.9 and 2.3.10), they are incomprehensible (consult a chromatography expert).
Response 5: We had rewritten the HPLC part.
6) Please explain the difference in the TPC content in the pear ferment and the sample at zero time for GF and/or IF (Figure 1)?
Response 6: It was explained as ‘According to the study, the TPC in green tea was significantly reduced in the initial phase of gastric digestion by a low-acid environment, and it was suggested that the reduction in TPC may be related to the instability of macromolecular phenolic components upon entering a low-acid environment, suggesting that pH plays an important role in altering the content of bioactive compounds [42]’ in discussion.
7) Present the results (TPC, TFC, SOD, protease activity, etc.) to the number of valid places corresponding to the precission of method! Presentation of results of TPC or TFC at hundredths of mg/mL is nonsense with respect to LOD.
Response 7: The linear range had been added and the number of valid had been adjusted.
8) The content of TPC in the GF2 sample (expressed as gallic acid) is lower (ca. 18 mg/mL, Fig. 1) than the content of gallic acid in the same sample determined by the HPLC method (ca. 75 mg/mL, Fig. 9b). Please explain.
Response 8: Sorry, we checked the original data and found that the determination method and experimental batch of gallic acid were different from other phenolic acids, which was not faithful and comparable. Therefore, we finally deleted the data of gallic acid.
9) Is the content of acetic acid in the range of 130-170 mg/mL realistic (Fig. 10)? Aren't the wrong units listed as milligrams instead of micrograms?
Response 9: It’s our mistake, the unit should be milligrams. We had corrected the data.
Best regards
Xingying Zhang

Reviewer 2 Report
Dear authors,
Your manuscript has potential to be interesting for researchers which analyse bioaccessibility of phenolics and antioxidant potential of different food (fruit) matrix after in vitro digestion. Further, your study also promote fermented fruit as a new functional product. However, there are some corrections/improvements and suggestions that must be done before any further Manuscript processing.
I will enumerate different requirements to improve the manuscript. All my comments are listed below and separated as General and specific comments.
General comments:
1. I suggest to you that meaningfully rewrite the existing version of the abstract and to point out the specifics of this research. The abstract must contain only key data from the whole manuscript.
2. The aim of this study is not appropriately written. Sentence (Line 62-67) is too confusing and major remarks of this study are not highlighted in an appropriate form.
“Your aim in this study was determined bioaccessibility and changes of bioactive compounds, as well as antioxidant properties of 'Xuehua' pear ferment after in vitro gastrointestinal digestion.”
3. Names of bacteria and yeast strains, as well as term "in vitro" should be written in Italic style. Please, check and correct this in the whole manuscript.
4. In the "Material and methods" section, give a detailed description of the methodology of in vitro gastrointestinal digestion (subsection 2.3.3.; Line 116). Is the oral phase of digestion skipped?
5. In section "Results" there are many terminological errors. I exhaustively listed all terminological errors in "specific comments" and I suggest to author that correct it.
6. I suggest to you that determine bioaccessibility (total recovery, %) of individually and total phenolic compounds at the end of in vitro gastrointestinal digestion (section 3.6).
7. Why are the results of control digestions without enzymes not included in the "results" section? It would be very interesting to compare the results of digestion with and without enzymes. So, I suggest to you that you add the obtained results for control digestion in the "Results" section.
Specific comments:
Line 15-19: This sentence ("The result showed … content decreased (p<0.05).") is too long, which makes it very confused. So, it must be clarified and meaningfully rewritten.
Line 38-52: This part of the introduction (These sentences) must be uniform and meaningfully rewritten. In the introduction, only a brief overview of the previous research is needed, it is not desirable to recount what was done in the research of other authors. The results from these researches (Line 38-52), you can use for comparison with your results in section (results and discussion).
Line 54-55: This sentence (“It contains abundant … flavonoids, polyphenols, etc.”) must be rewritten. Vitamins are bioactive compounds, while flavonoids are subclasses of phenolic compounds. Please, correct it.
Line 55: This sentence (“Therefore, studies on the … digestion are necessary.”) was misinterpreted and must be meaningfully rewritten.
Line 132: This sentence (“The Folin-Ciocalteu method … slight modifications”), must be meaningfully rewritten. Please, correct it.
Line 143: According to the method for the determination of total flavonoids, 10% aluminium chloride is most often used. Why do you use aluminium nitrate?
Line 183: This sentence (“The method of the … slightly modified”)must be meaningfully rewritten. Please, correct it.
Line 191: This sentence (“The method … reference [30].”), must be meaningfully rewritten. Please, correct it.
Line 210-212: This sentence (“The method performed … with slight modifications”) must be meaningfully rewritten.
Line 212: Term “passes” replace with term “filtered”. Term “with” delete.
Line 223-224: This sentence (“The method was reference … modifications.”), must be rewritten.
Line 240: “The TPC of pear ferment decreased after gastric and intestinal digestion, in comparison to the initial sample of pear ferment.” This must be emphasized, so as not to confuse the reader.
Line 243-244: “… when sample transferred into the simulated intestinal environment, but …”. Please, correct according to suggestion.
Line 257: You can write this sentence as follows: ("There was no significant change in TFC after transferred of sample from gastric phase to intestinal phase"), to make it clearer.
Line 267: “… The protease activity of initial pear ferment sample was significantly increased after gastrointestinal digestion …”. Please, correct according to suggestion
Line 270: “…when transferred into the intestinal fluid…”. Please, correct according to suggestion.
Line 272: “…and then the same constant activity, without significant difference, until the end of the gastric phase (p>0.05),…”. Please, correct according to suggestion.
Line 280-281: “The SOD activity increased during gastric digestion and decreased during intestinal digestion.”. Please, correct according to suggestion.
Line 284: “… when transferred into the…”. Please, correct according to suggestion.
Line 302: “… within 0.5h (p<0.05), and activity remains constant thereafter (p<0.05)…”
Line 310: “In Fig. 6 showed the changes in ·OH-RSA of ‘Xuehua’ pear ferment, during in vitro gastrointestinal digestion.”. Please, correct according to suggestion.
Line 311: Term “extremely” delete. It is not appropriate in this context.
Line 314: Term “changed into the intestinal digestion” replace with term “transferred into the intestinal phase of digestion”
Line 326: “… when the sample transferred to simulated intestinal environment.” Please, correct according to suggestion.
Line 336: Term “total phenols” replace with term “total phenolics”
Line 336-339: This sentence (“To further investigate … performed in ‘Xuehua’ pear ferment.”) is confused and must be rewritten.
Line 349: “(p<0.05).” Replace with “(p<0.05),”
Line 362: Gibberellin is not a flavonoid. Please delete "gibberellin".
Line 366: This sentence (“When changed to the … of both increased”) must be rewritten.
Line 369: This part of sentence “… but Untested at 2 h of digestion.”) is confused. Please check and correct it.
Line 372: Term “And the” at the start of sentence delete
Line 372: Which phenolic compounds ?
Line 375: “When transferred to intestinal …”
Line 376: This part of sentence “….than the final content of GF…” replace with “… in comparison to their the final content after GF …”.
Line 380: Term “level” replace with term “content”
Line 381: “… while protocatechuic acid was not detected after 2h of digestion.” Please, correct according to suggestion.
Line 382 and 384: Term “level” replace with term “content”
Line 397: This part of sentence “… HPLC evaluation identified…‘Xuehua’ pear ferment.”) is confused and must be rewritten.
Line 400: “When transferred to simulated gastric environment…”
Line 410: “When transferred to simulated intestinal environment, the content of organic acids decreased, in comparison to their final conten after gastric phase,…”. Please, correct according to suggestion.
Line 434-438: This sentence (“In this study, … of ‘Xuehua’ pear ferment.”)is confused, and must be meaningfully rewritten.
Line 439: Term “total phenols” replace with term “total phenolics”. Please, check in whole manuscript and correct it.
Line 475-477: This sentence (“This indicated that … of flavonoids as arbutin”) indicates some statements that are not true, such as “pepsin to decompose the flavonoids”… Pepsin is exclusively a proteolytic enzyme!!! So, this sentence must be meaningfully rewritten.
Line 498-501: This sentence (“While there’s a … enzymatic reaction rate”) is confused and must be rewritten.
Line 506: “… when sample transferred into the intestinal…”. Please, correct as I suggested you.
Line 558:”… when sample transferred into the simulated intestinal environment.”. Please, correct according to suggestion.
Line 569-574: This sentence (“which was also confirmed … the antioxidant activity [56].”) is confused and too long. So, it must be rewritten.
Author Response
Dear Reviewer,
Thank you for your valuable comments.
We carefully read the comments of all reviewers and revised the article as follows:
- Some parts of method, result and discussion were rewritten to make the description more clear and accurate.
- The data of digestive control group was added as required. Some inaccurate data was corrected.
- We enhanced English expression and modified some confusing sentences.
We put the manuscript in word with ‘track change’ function to show where we revised. Please see the attachment.
Here I would like to answer your questions and suggestions on this paper.
General comments:
- I suggest to you that meaningfully rewrite the existing version of the abstract and to point out the specifics of this research. The abstract must contain only key data from the whole manuscript.
Response 1: The abstract had been rewritten.
- The aim of this study is not appropriately written. Sentence (Line 62-67) is too confusing and major remarks of this study are not highlighted in an appropriate form.
“Your aim in this study was determined bioaccessibility and changes of bioactive compounds, as well as antioxidant properties of 'Xuehua' pear ferment after in vitro gastrointestinal digestion.”
Response 2: We have rewritten the sentence as ‘Therefore, it is necessary to study the bioaccessibility and changes of the bioactive substances, as well as the antioxidant properties in ‘Xuehua’ pear ferment during simulated gastrointestinal digestion.’
- Names of bacteria and yeast strains, as well as term "in vitro" should be written in Italic style. Please, check and correct this in the whole manuscript.
Response 3: We had revised the words form in the whole manuscript that needs italics.
- In the "Material and methods" section, give a detailed description of the methodology of in vitro gastrointestinal digestion (subsection 2.3.3.; Line 116). Is the oral phase of digestion skipped?
Response 4: Only simulated gastric fluid digestion and simulated pancreatic digestion were done in the study design, and oral digestion was not designed.
- In section "Results" there are many terminological errors. I exhaustively listed all terminological errors in "specific comments" and I suggest to author that correct it.
Response 5: We revised the manuscript as suggested in "specific comments".
- I suggest to you that determine bioaccessibility (total recovery, %) of individually and total phenolic compounds at the end of in vitro gastrointestinal digestion (section 3.6).
Response 6: We had added a figure of bioaccessibility of phenolic compounds in section 3.6.
- Why are the results of control digestions without enzymes not included in the "results" section? It would be very interesting to compare the results of digestion with and without enzymes. So, I suggest to you that you add the obtained results for control digestion in the "Results" section.
Response 7: We had added the control group as gastric control (GC) and intestinal control (IC) into the result.
Specific comments:
We revised the terminological errors and confusing sentences in the manuscript as suggested. Here we would like to explain some of questions.
Line 38-52: This part of the introduction (These sentences) must be uniform and meaningfully rewritten. In the introduction, only a brief overview of the previous research is needed, it is not desirable to recount what was done in the research of other authors. The results from these researches (Line 38-52), you can use for comparison with your results in section (results and discussion).
Response: We condensed the discussion in this section as “Previous studies had showed that the bioavailability of phenolic compounds in fruits was greatly reduced after the simulated gastrointestinal digestion especially during intestinal digestion [13-14]. In contrast, there were researches found that simulated gastrointestinal digestion increased the bioavailability of phenolic compounds actives and antioxidant activity in different kinds of fruit ferment [15-18].
Line 143: According to the method for the determination of total flavonoids, 10% aluminium chloride is most often used. Why do you use aluminium nitrate?
Response: Aluminum nitrate colorimetric method is commonly used in Chinese studies. We put a new reference using aluminum nitrate in TFC method.
Best regards
Xiaoying Zhang

Round 2
Reviewer 1 Report
Dear author(s),
I have read your response to my questions and I agree with them, with two exceptions.
1) The HPLC methodology is written (even after the correction of the first version) in a style that does not correspond to the common one. However, it is already more understandable.
2) The units of acetic acid concentration given in the text (units of mg/L) do not correspond with those given in Figure 10 (hundreds of mg/L).
I found several ambiguities in the revised manuscript v2.
See the notes/comments in the attached file foods-2196285-peer-review-v2_rev.pdf and edit the manuscript.

Author Response
Dear reviewer,
Thank you for your valuable comments.
We had carefully read all the comments and made the following revisions:
- We carefully modified the description of the method part to make it more accurate.
- We checked the previously changed data and adjusted the figure according to the data.
Here I would like to response to your comments and questions.
- The HPLC methodology is written (even after the correction of the first version) in a style that does not correspond to the common one. However, it is already more understandable.
Response: Thank you for your patient guidance. The writing style of HPLC methodology referred to Su, et al. (Su, et al. The effect of simulated digestion on the composition of phenolic compounds and antioxidant activities in lychee pulp of different cultivars. International Journal of Food Science & Technology 2019, 54, 3042-3050. http://doi.org/10.1111/ijfs.14217) and Quinatzin, et al. (Quinatzin, et al. Organic Acids, Antioxidants, and Dietary Fiber of Mexican Blackberry (Rubus fruticosus) Residues cv. Tupy[J]. Journal of Food Quality, 2018, 2018:1-9.). We retained the description format and wording style of two references to be more scientific. This time we further modified the description according to the published article as follow from Foods. (Zhang, et al. Changes in Phenolic Compounds and Antioxidant Activity during Development of ‘Qiangcuili’ and ‘Cuihongli’ Fruit. Foods 2022, 11, 3198. https://doi.org/10.3390/foods11203198).
- The units of acetic acid concentration given in the text (units of mg/L) do not correspond with those given in Figure 10 (hundreds of mg/L).
Response: We had checked the data and didn’t find the mistake. Actually the figures carried with the original wrong data were showed together with the revision figure under“Track Changes” mode. We think it made confusion with viewing the two vision of figures at the same time. We had recreated the figures with clearer coordinates. Please check the new version.
We also corrected the ambiguities according to your attached file.
We put the revised manuscript in MS Word with ‘track change’ function to show where we revised. Please see the attachment.
Thank you again for your kind support.
With best regards.
Yours sincerely,
Author team of foods-2196285

Reviewer 2 Report
I have no any added comments.
Author Response
Dear reviewer,
We had carefully read the comments from all reviewers and made the following revisions:
- We carefully modified the description of the method part to make it more accurate.
- We checked the previously changed data and adjusted the figure according to the data.
We put the revised manuscript in MS Word with ‘track change’ function to show where we revised. Please see the attachment.
Thank you again for your kind support.
With best regards.
Yours sincerely,
Author team of foods-2196285
